# Mechanical and Durability Properties of Aerated Concrete Incorporating Rice Husk Ash (RHA) as Partial Replacement of Cement

**Tariq Ali [1], Abdullah Saand [1], Daddan Khan Bangwar [1,*], Abdul Salam Buller [2] and Zaheer Ahmed [3]**

1   Civil Engineering Department, Quaid-e-Awam University of Engineering, Science & Technology, Nawabshah 67450, Pakistan; tariqdehraj@gmail.com (T.A.); abdullah@quest.edu.pk (A.S.)
2   Civil Engineering Department, Larkana Campus, Quaid-e-Awam University of Engineering, Science & Technology, Larkana 77150, Pakistan; buller.salam@quest.edu.pk
3   Civil Engineering Department, College of Engineering & Technology, University of Sargodha, Sargodha 40100, Pakistan; zaheer.ahmed@uos.edu.pk
*   Correspondence: daddan@quest.edu.pk

**Abstract:** In today's world, the implementation of industrial ecology for sustainable industrial development is a common practice in the field of engineering. This practice promotes the recycling of by-product wastes. One of those by-product wastes is rice husk ash. This paper describes an investigation into the effect of rice husk ash (RHA) as a partial replacement for cement, to produce lightweight, aerated concrete. Type I Portland cement, fine aggregate, and aluminum powder as an aerating agent were used in this study. The RHA was used in different replacement levels, i.e., RHA was used to replace cement at 0%, 2.5%, 5%, 7.5%, 10%, 12.5% and 15% by weight. Aluminum powder was added during mixing at 0.5% by weight of binder to obtain lightweight, aerated concrete. Test results are presented in terms of physical, mechanical, and durability aspects that include density, compressive strength, split tensile strength, and flexural strength of concrete cured at different curing regimes, i.e., 3, 7, 28, and 90 days along with corrosion analysis, and sulphate attack at 28 days of curing. The test results show that using 10% RHA as a partial replacement of cement in aerated concrete is beneficial in triggering the strength and durability properties of concrete.

**Keywords:** rice husk ash (RHA); aluminum powder; aerated concrete; mechanical properties; durability properties

## 1. Introduction

Nowadays, aerated concrete is used in many construction applications because of its lightweight. Lightweight concrete structures are easy to cast and build, and are also easy to maintain. In civil engineering applications, lightweight structural concrete is often used as a very easy alternative to traditional concrete. Lightweight concrete decreases the total self-weight of the buildings, leading to a decrease in the size of the foundations, thereby lowering costs and improving durability.

Compared to traditional concrete, aerated concrete is a relatively homogeneous material because it lacks the coarse aggregate, yet it exhibits wide variation in its characteristics. Aerated concrete is created using either chemical admixtures or mechanical foaming agents by trapping air in the matrix. This makes the product lightweight and accounts for most of its beneficial effects, namely material savings, strong thermal insulation, and benefits that lead to economic savings [1]. The significant amount of air voids inside the concrete matrix produced with the help of some aerating agents without using coarse aggregates is classified as aerated concrete [2].

Depending on the curing process, aerated concrete can be autoclaved or non-autoclaved. The aerated concrete cured under the steam curing regime is known as autoclaved aerated concrete (AAC). The absorption properties, drying shrinkage, compressive strength, etc.,

depend on the curing technique and time [3]. Autoclaved aerated concrete is indeed a lightweight concrete typically used in building applications. Compared to traditional construction materials, AAC gives reduced thermal conductivity, reduced density, resistance to fire, lower shrinkage, and simplicity of usage in the building industry. Ropelewski & Neufeld [4] developed AAC samples from the electric utility fly ash as the silica source and thermal inertial properties were evaluated. It was suggested that the theoretical periodic heat flow model in the literature for thermal inertia did an adequate job of predicting the observed thermal inertia parameters for the AAC and conventional construction samples. Narayanan & Ramamurthy [5] summarized the work carried out to investigate the performance of aerated concrete. The focus of this paper is to classify the investigations into the properties of aerated concrete in terms of physical, mechanical, and chemical properties. Dey et al. [6] determined the impact flexural response of aerated concrete using Three-point bending test (3PBT); the effect of impact energy on the mechanical properties was investigated and they concluded that dynamic flexural strength under impact was more than 1.5 times higher than static flexural strength.

The aluminum reacts with calcium hydroxide or alkali during the slurry process to create hydrogen gas bubbles that lead to an increase in the aerated concrete porosity [7].

A growing number of articles have been published in recent times regarding the manufacture of concrete by the usage of residues from agricultural, industrial, and thermoelectric plants. In concrete, a significant role has been played by different materials with pozzolanic properties, like fly ash, rice husk ash, and silica fume. The consumption of mineral admixture by cement increased in the late 20th century and concrete is now achieved by the partial replacement of cement [8,9]. Rice husk ash is used as a partial cement replacement material; the major utilization of rice husk in rice paddy milling is as fuel. However, the use of such a fuel creates an immense amount of ash that has no immediate significant benefit and is typically poured into water streams causing waste and spring contamination. As a consequence, the use of RHA has developed great interest [3]. A very common agricultural by-product in different regions of the globe is the rice husk, which is the outer shell of a rice grain, produced by rice milling. Raw rice husk (RRH) comprises approximately 20% silica group, 30% lignin group, and 40% cellulose. This RRH is usually used in the rice milling industry as a fuel in the parboiling procedure. During combustion, the RRH cellulose–lignin burns and tends to leave behind a porous silica skeleton. Consequently, RHA holds silica in a high amount.

A fine powder with a high surface area is produced after grinding the porous silica skeleton of rice husk, named rice husk ash (RHA). RHA is seen as a highly reactive pozzolanic material in the manufacture of concrete owing to its large silica content. The reactivity of rice husk ash is responsible for the large amorphous silica percentage and the high surface area control by the particle porous structure [2,10–12]. Many mineral admixtures have a beneficial effect on the durability and strength of concrete. In the case of RHA, its chemical impact is linked to the fact that it would be a highly pozzolanic material that rapidly combines calcium hydroxide to form a secondary C–S–H when developed by controlled combustion [3]. The impact of rice husk ash on autoclaved aerated concrete's physical and mechanical properties was investigated for 8 h and 18 h, autoclave curing was carried out at 180 °C. Based on the outcome, the unit weight decreased in rice husk concrete (RHC) as a result of the comparatively low specific gravity, i.e., 2.13 and greater RHA pore structure comparison to quartz sand, i.e., 2.59. RHC's thermal conductivity also declined as the RHA dose increased and RHC's compressive strength declined with the growing substitution of quartz sand with RHA, but it remained unchanged or only slightly altered with the difference in the autoclaving time, [7]. The impact of rice husk ash on aerated concrete was also observed and two distinct types of curing were used, i.e., steam curing and moist curing. The tests for unit weight and compressive strength were calculated at the different replacement levels, the RHA was replaced with sand with the addition of aluminium powder at 0.5 percent by weight. Findings demonstrated that, with increased RHA material, the unit weight decreased so it was almost constant or slightly under during

different curing conditions and periods. The unit weight has a direct relationship with compressive strength: a lower unit weight results in a decrease in compression strength. It was also noticed that when RHA is used the compressive strength decreases. This is because the RHA structure is extremely porous and thus needs additional water in the mix to maintain the aerated concrete's workability throughout molding [13]. In their analysis, Kartini et al. [14] concluded that greater percentages of RHA substitution result in a decrease in compressive strength. However, using RHA instead of cement for 10% of the cement replacement achieved the desired compressive strength; substitution of RHA not only improved the compressive strength but also the durability properties enhanced. Another researcher found that the substitution of RHA for cement in concrete increased the amount of water needed and at 10% replacement RHA concrete provided a significant increase in strength [15]. According to the findings, the highest RHA substitute for cement in this study was 5%, which offered the highest compressive strength at 28 days [16].

The use of rice husk ash as a partial substitute for cement in conventional concrete has been extensively researched in terms of mechanical strength, physical properties, and durability. But limited research work has so far been conducted on the utilization of rice husk ash as a cement replacement in the production of aerated concrete.

Therefore, this study aimed to reduce manufacturing costs and protect the environment from the waste's adverse effects. The objective of the present research was to investigate the effect of using rice husk ash (RHA) as a partial replacement for cement up to 15% to produce lightweight, aerated concrete on the physical, mechanical, and durability properties, i.e., density, compressive strength, split tensile strength, flexural strength, corrosion potential, and sulphate attack. The materials chosen were carefully studied concerning their properties such as X-ray Diffraction (XRD), Energy Dispersive Spectrometry (EDS), Scanning Electron Microscope (SEM), and specific gravity. All specimens were cured at different curing regimes, i.e., 3, 7, 28, and 90 days.

## 2. Experimental Campaign

### 2.1. Materials and Mix Proportions

Ordinary portland cement, quartz sand, RHA, and water were developed to blend aerated concrete. The RHA included in this research was synthesized for 2 h at a temperature of 800 °C in a laboratory using an electric furnace. Then it was ground by American society for testing and materials (ASTM C618) before being passed through sieve no. 325. The RHA was used as a partial cement substitution. Quartz sand was also sieved and passed through sieve no. 30 or 600 microns. Aluminium powder was used as an aerating agent at a dosage of 0.5% by weight of the binder. Energy Dispersive Spectrometry (EDS) was used to identify the chemical compositions of the RHA used for this study, and the outcomes are listed in Table 1. The physical characteristics of these materials are described in Table 1. The results verify that the material is largely composed of $SiO_2$. Furthermore, Table 1 shows the presence of $SiO_2$ is in the largest quantity of the basic elements in the extracted RHA. Such materials are deemed to be one of the pozzolanic materials in nature. The microstructure of rice husk ash was obtained using a Scanning Electron Microscope (SEM). Scanning electronics microscopy snaps taken at different resolutions show that the optimized sample RHA (800 °C—2 h) is multifaceted with angular, micro-absorbent surfaces, the images also indicate its high specific surface, as shown in Figures 1 and 2, respectively. The mineralogical characteristics of RHA was determined qualitatively by X-Ray diffraction (XRD) as shown in Figure 3. A diffractometer system was used to scan the material. An X-ray diffractometer was used to look into the samples of RHA carefully using Cu $K\alpha$, radiation at 40 kV/30 mA, CPS = 1 k, width 2.5, speed 2°/min, and scanning from 2θ range = 10–80°. The results in Figure 3 show that the XRD patterns of RHA (800 °C—2 h), having a broad peak at an angle 2θ of 19°, confirmed that the material contains an amorphous silica structure. In this research, binder/sand ratio (1:1) of 50:50% by weight was used. RHA was used to replace part of ordinary Portland cement at 2.5–15% with an increment of 2.5% by weight. RHA mix proportions for this research are given in Table 2.

**Table 1.** Properties of OPC and optimized Rice Husk Ash (RHA).

| Material | Physical Properties | | Chemical Analysis (% age) | | | | | | |
|---|---|---|---|---|---|---|---|---|---|
| | Blaine (cm²/g) | Specific Gravity | $SiO_2$ | CaO | $Al_2O_3$ | MgO | $K_2O$ | $Fe_2O_3$ | LOI |
| **Cement** | 3008 | 3.15 | 20.78 | 60.89 | 5.11 | 3.00 | 0.00 | 3.17 | 1.71 |
| **RHA** | 2251 | 2.12 | 74.35 | 1.39 | 1.379 | 1.06 | 3.51 | 1.029 | 1.50 |

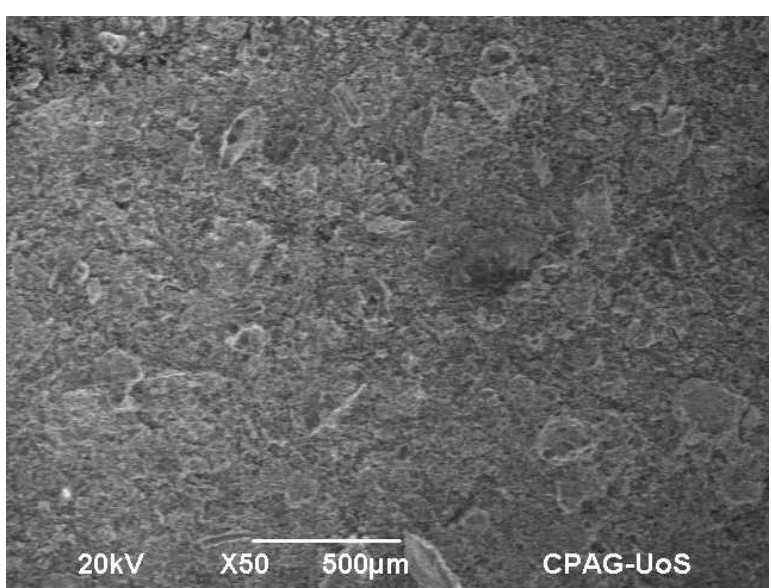

**Figure 1.** Scanning Electron Microscope (SEM) image at ×50 maginification (20 kV; 500 μm; 800 °C—2 h) of RHA particles.

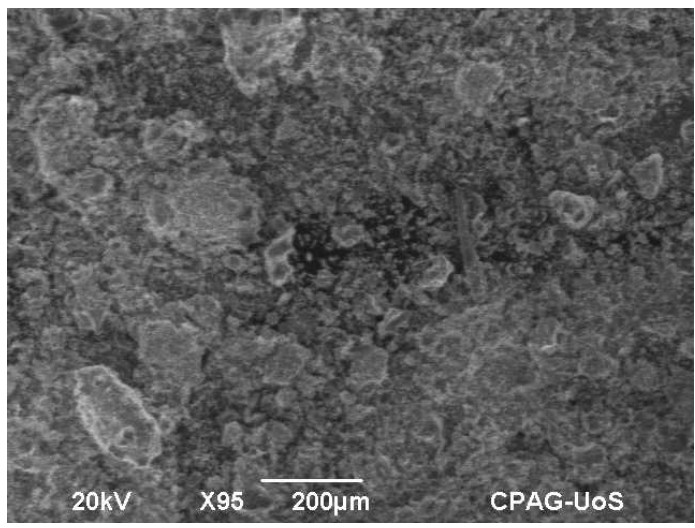

**Figure 2.** Scanning Electron Microscopy (SEM) image at ×95 maginification (20 kV; 200 μm; 800 °C—2 h) of RHA particles.

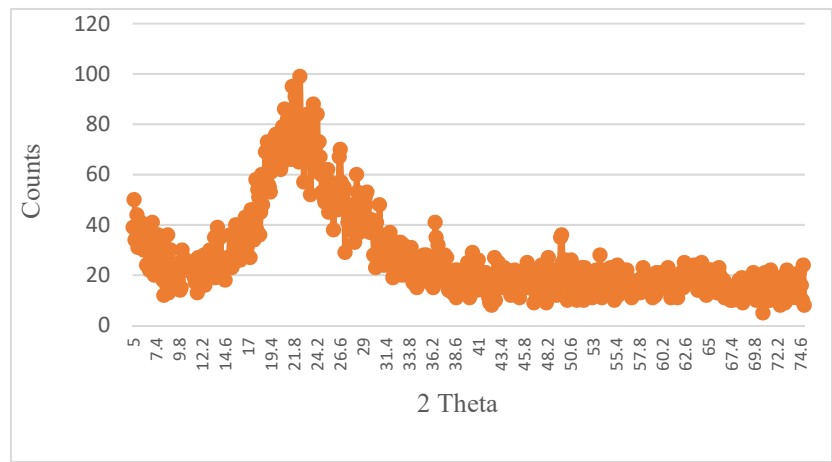

**Figure 3.** X-ray Diffraction (XRD) pattern of RHA (800 °C—2 h).

**Table 2.** Mixture proportion of control mix aerated concrete and rice husk ash aerated Concrete.

| S.No | Concrete Mix | Ratio | RHA (%) | Cement (kg) | RHA (kg) | W/C | W/B | Water (kg) | F.A (kg) | Aluminum Powder (%) | Sample Per Batch |
|---|---|---|---|---|---|---|---|---|---|---|---|
| 1 | M1 | 1:1 | 0 | 10 | 0 | 0.60 | 0.6 | 6 | 10 | 0.5 | 60 |
| 2 | M2 | 1:1 | 2.5 | 9.75 | 0.25 | 0.60 | 0.61 | 6.1 | 10 | 0.5 | 60 |
| 3 | M3 | 1:1 | 5 | 9.5 | 0.5 | 0.60 | 0.62 | 6.2 | 10 | 0.5 | 60 |
| 4 | M4 | 1:1 | 7.5 | 9.25 | 0.75 | 0.60 | 0.63 | 6.3 | 10 | 0.5 | 60 |
| 5 | M5 | 1:1 | 10 | 9.0 | 1.0 | 0.60 | 0.64 | 6.4 | 10 | 0.5 | 60 |
| 6 | M6 | 1:1 | 12.5 | 8.75 | 1.25 | 0.60 | 0.65 | 6.5 | 10 | 0.5 | 60 |
| 7 | M7 | 1:1 | 15 | 8.5 | 1.5 | 0.60 | 0.66 | 6.6 | 10 | 0.5 | 60 |

W/C = water/cement; W/B = water/binder; S. No = serial number; F.A = fine aggregate; RHA = rice husk ash.

### 2.2. Specimen Preparation

The laboratory pan mixer was used; before casting the specimens the materials were sieved. Batching (by weight) was performed to have the appropriate quantities of the ingredients in the required mix, the cement to sand ratio used in the research was 1:1. Throughout the research work, steel molds were used which were cleaned and assembled properly.

The inner surfaces of the molds were oiled to avoid the sticking of concrete to the mold surfaces and to achieve easy de-molding. The casting was performed by standards. Molds were de-molded after 24 h of the casting and the specimens were properly marked following the curing with the testing age. Entire specimens were cured for moist curing at different curing regimes, i.e., 3, 7, 28, and 90 days.

### 2.3. Test Methods

For the development of rice husk ash aerated concrete, the following tests were carried out: density, compressive strength, split tensile strength, flexural strength, corrosion potential, and sulphate attack. For compressive strength, test cube specimens were cast with dimensions of 100 mm × 100 mm × 100 mm according to American society of testing and materials (ASTM C1693). For split tensile strength, the cylinder specimens of size 100 mm × 200 mm were cast according to ASTM C 496. To perform the flexural strength test the prism specimens of size 40 mm × 40 mm × 160 mm were cast according to ASTM C 293/C293M-16. A Universal Testing Machine (UTM), with a capacity of 1,800,000 N, was used to perform the mechanical tests. Unit weight/density were also calculated before being subjected to final testing. The dimensions of each specimen are shown in Figure 4.

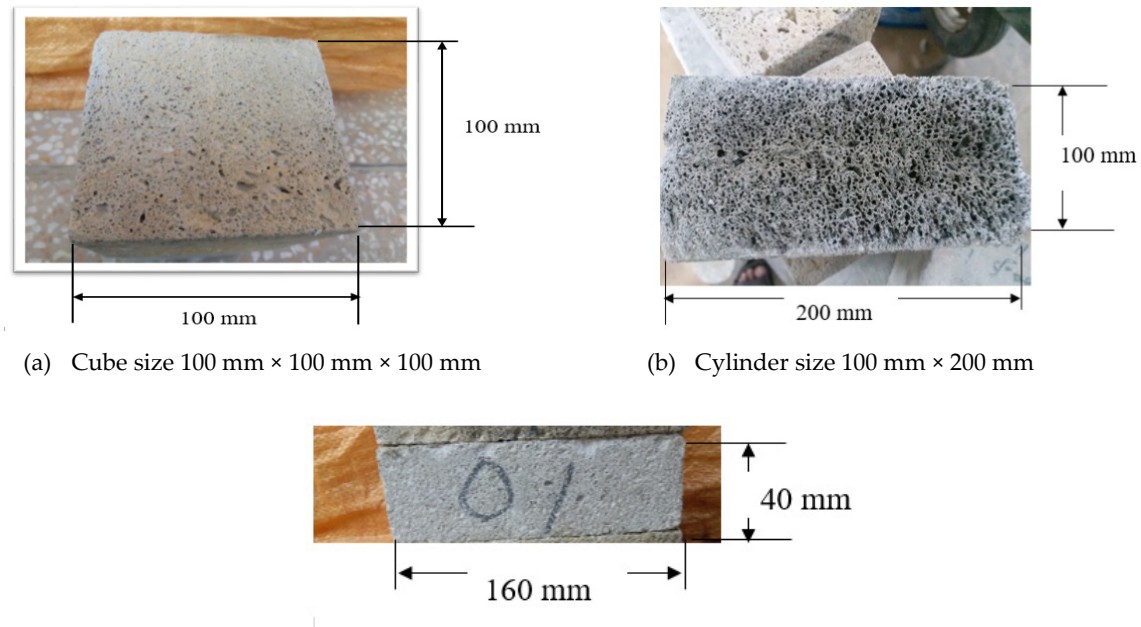

(a)  Cube size 100 mm × 100 mm × 100 mm

(b)  Cylinder size 100 mm × 200 mm

(c)  Prism size 40 mm × 40 mm × 160 mm

**Figure 4.** Dimension of specimens tested during this study. (**a**) dimension of cube specimen (**b**) dimension of cylindrical specimen (**c**) dimension of prism specimen.

For corrosion analysis, cylinder specimens of size 100 mm × 200 mm were cast. In the center of all the specimens for corrosion analysis tests, a bar of 12 mm diameter, 300 mm in length was mounted. After 24 h, samples were de-molded and placed for moist curing for 28 days. The samples were then placed in a water tank containing a solution of 3% sodium chloride for 15 days after the completion of the curing process. After 15 days of immersion in sodium chloride solution, samples were taken out and placed in the open air for air drying curing for 15 days. The same cycle of moist and air curing was repeated for up to 90 days. According to ASTM C-876, corrosion potential was assessed. The mean values of 5 samples were reported.

The Sulphate resistance of concrete was assessed by determining the length variations of specimens that were prepared in the shape of mortar bars with a size of (25 mm × 25 mm × 285 mm) by ASTM C1012. After 24 h of casting, prisms were de-molded and then put in moist curing for 28 days. After the completion of curing, all samples were kept in the sodium Sulphate solution ($Na_2SO_4$). Change in length was recorded weekly with the help of a digital meter. Figure 5 shows the casted and fractured specimens before and after testing.

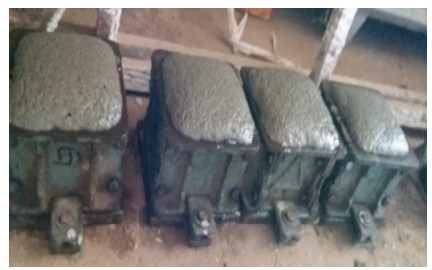

(a)　Cube specimen during casting

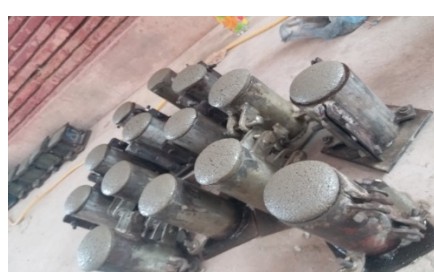

(b)　Cylindrical specimen during casting

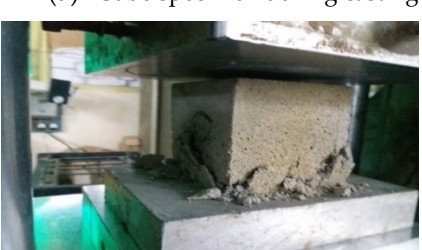

(c)　Cube specimen during testing

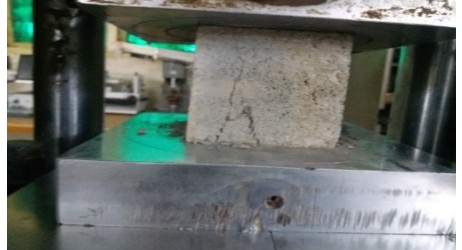

(d)　Fractured cube specimen

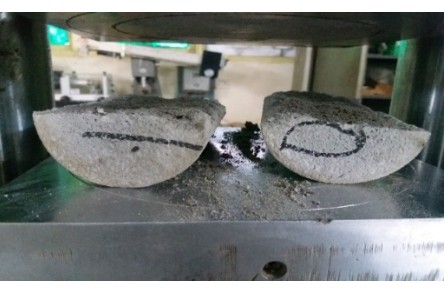

(e)　Fractured cylindrical specimen

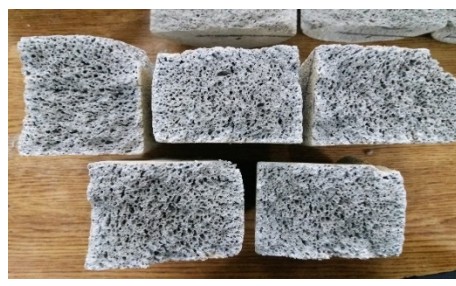

(f)　Fractured prism specimen

**Figure 5.** Fabricated and fracture images of all specimens (**a–f**).

## 3. Results and Discussion

### 3.1. Density

The density of RHA samples, cured for moist curing at various curing regimes, i.e., 3, 7, 28 and 90 days, are presented in Figure 6. As RHA is added, the unit weight of aerated concrete with RHA is increased, because as the percentage of RHA increases in the matrix it starts affecting the aeration process, which results in an increase in density of up to 10% replacement with cement. Also, it was noted that the essential hydration and reaction products for the RHA paste were calcium hydroxide [$Ca(OH)_2$] and calcium silicate hydrates [C–S–H]. The paste incorporating RHA had decreased $Ca(OH)_2$ content compared with that of the control Portland cement paste due to pozzolanic reaction. The integration of the RHA in concrete decreased its porosity [17]. RHA reduces the porosity and width of the interfacial zone in a way that increases the density [18]. Reduction in the porosity increases the unit weight of the AAC developed [7]. Moreover, at 12.5% and 15% replacement of RHA the density starts decreasing due to the specific gravity difference of cement and RHA as shown in Table 1. This clearly shows that the particles of cement are heavier compared to rice husk ash particles which helps in reducing the density. Studies indicate that, owing to the porous structure and lower specific gravity, the unit weight decreases with increased RHA percentage [7,13].

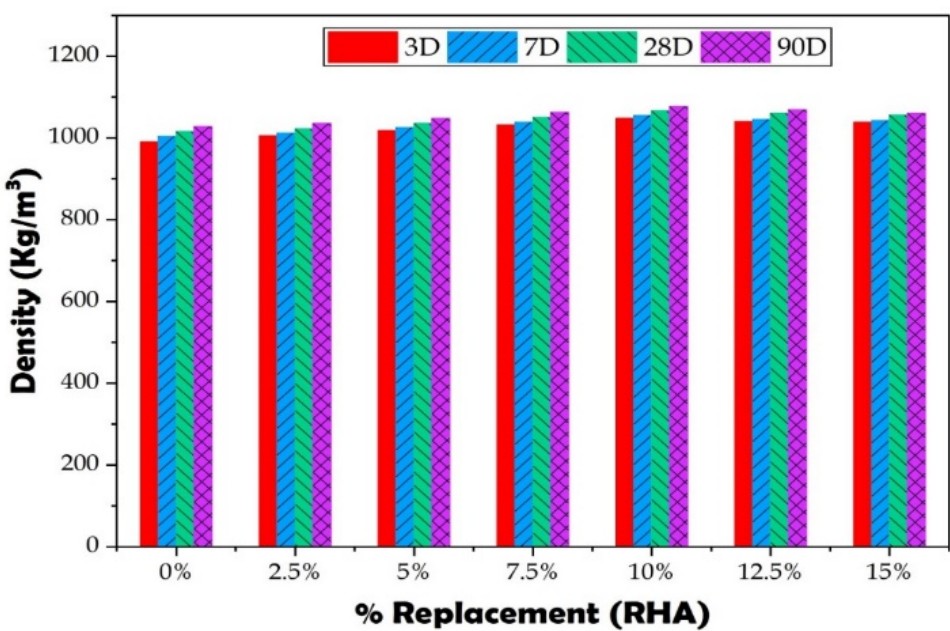

**Figure 6.** The density of aerated concrete (D—days).

### 3.2. Mechanical Properties

The compressive strength, split tensile strength, and flexural strength test results of RHA samples are presented in Figures 7–9. The samples were cured for moist curing at various curing regimes, i.e., 3, 7, 28, and 90 days.

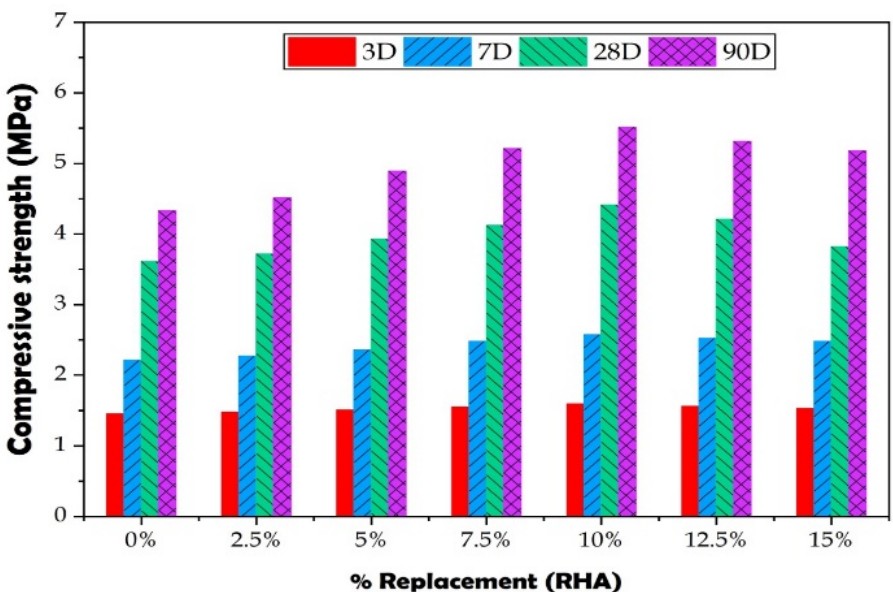

**Figure 7.** Compressive strength of aerated concrete (D—days).

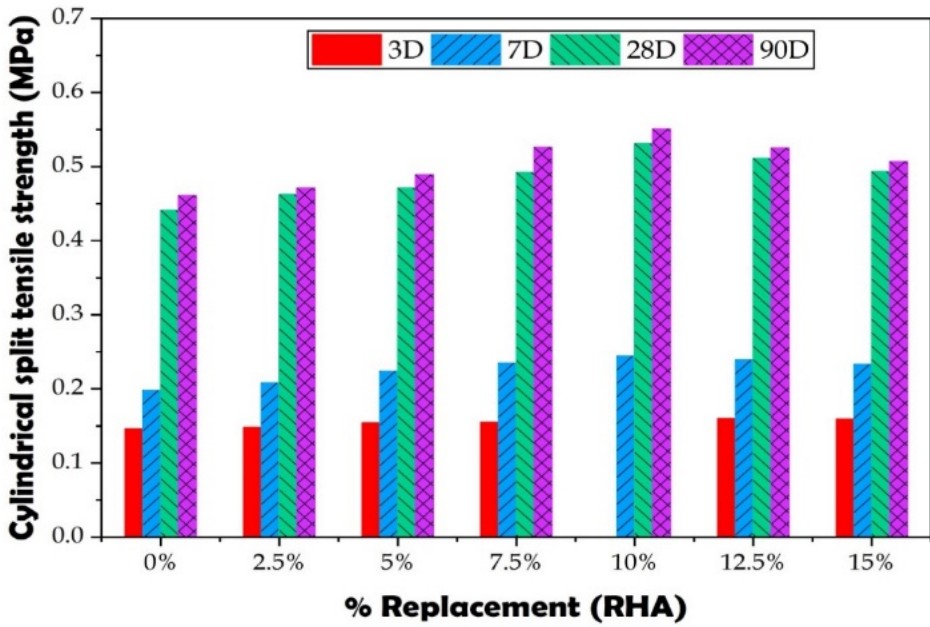

**Figure 8.** Split Tensile Strength of aerated concrete (D—days).

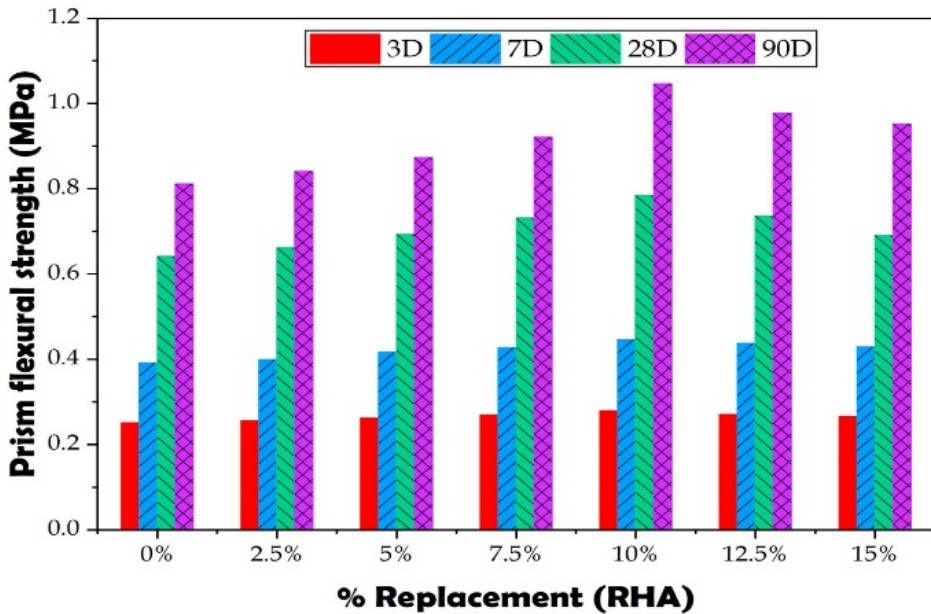

**Figure 9.** Flexural strength of aerated concrete.

The mechanical properties of aerated concrete with RHA are increased up to 10% RHA replacement with cement. It was observed in results that the gradual rise in strength is due to the increase in density. The increase in strength is because of the direct relationship between density and strength. Also, it has been seen that the increase in strength is due to the presence of reactive silica in RHA. The reactive silica in the RHA blended concrete paves the way for the formation of a strength-giving compound, i.e., secondary supplementary C–S–H gel. Such a trend in compressive strength was also reported by many researchers [19–25]. Moreover, as RHA percentage replacement increases up to 12.5% and 15%, a decreasing trend in strength is noticed. Due to the replacement of cement with RHA, the decrease in compression strength caused the water demand to increase, which adversely affected the compressive strength of the AAC [7].

Kunchariyakun et al. [13] observed that a reduction in compression strength is noticed with a decrease in unit weight, which shows the direct relationship between compressive strength and unit weight. The RHA structure is extremely porous and thus needs further water in the mix to retain the aerated concrete's functionality throughout molding. The earlier studies showed that the increased water–binder ratio contributes to reducing compression strength [26]. The compressive strength of RHA-containing concrete is related to water ratio, though at least a 10% cement replacement with RHA results in strength improvement comparable to control samples [27]. It is worth noting that with enhancing the RHA percentage more quantity of water is needed for the aeration process (formation of hydrogen gas) to take place. It is also shown that the decline in the strength of rice husk ash concrete is due to the high demand for water, such a trend is endorsed by Cook et al [28], and al Khalaf & Yousif [21]. Figure 10 depicts the fractured images of specimen during mechanical testing.

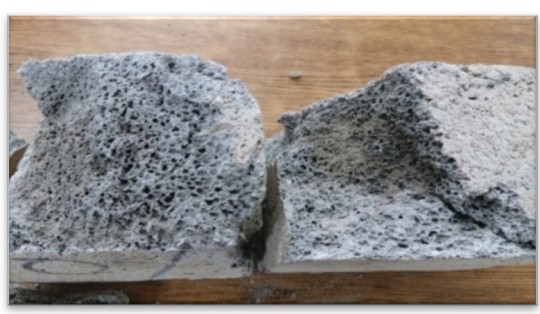

(a)　Cross-section of cube specimen

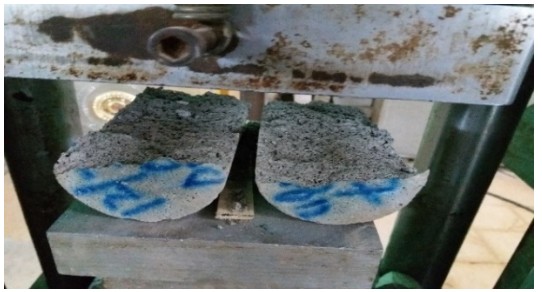

(b) Cross-section of prism specimen

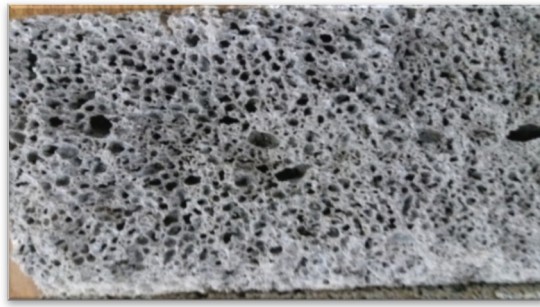

(c) Cross-section of cylinder specimen

(d) Cross-section of cylinder specimen

**Figure 10.** Cross-section images of fractured specimens (**a**–**d**).

### 3.3. Durability

3.3.1. Corrosion Analysis

The corrosion potential results at 28 days wet curing with 90 days sodium chloride curing for control and rice husk ash aerated concrete are shown in Figure 11. The results show that in the control mix 90% corrosion is active. Furthermore, when there are increments in the percentage of RHA in the aerated concrete matrix, the corrosion start decreases. The study reveals that the corrosion potential is decreased as the proportion of RHA is increased. RHA is very effective in resistance to the penetration of chloride. The reason behind this phenomenon is that the incorporation of pozzolans helps in the pore refinement of concrete's microstructure.

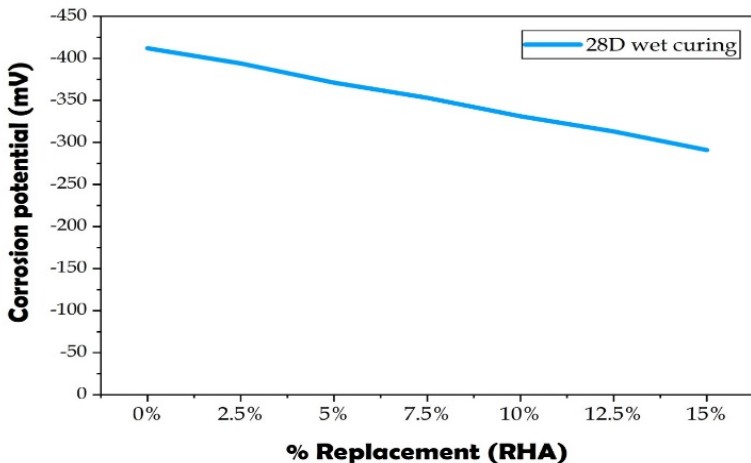

**Figure 11.** Corrosion potential (mV) of aerated concrete (D—days).

Consumption of calcium hydroxide by the formation of secondary C–S–H gel leads to the passivation (alkaline) environment in the concrete. According to Gemma Rodríguez de Sensale [3], the best resistance to chloride-ion penetration is achieved with a 15% substitution of RHA for Portland cement. It is also observed that when RHA is used in the concrete chloride there is ion penetration resistance [27].

### 3.3.2. Sulphate Attack

Sulphate attack is one of the most important properties of durability that can be influenced by the replacement of cement with rice husk ash. The expansion of the aerated concrete bar without RHA, i.e., control mix (CM1), is much larger than the samples produced with RHA. The results in Figure 12 show that with an increment of RHA dosage the samples resist the effects of sulphate attack. The same trend has been noticed by different researchers: RHA-containing specimens were found to be more resistant to HCl solution and sulphate attack than those without RHA [3]. Rice husk ash-developed concrete gives better resistance to sulphate attack [27]. Where M mentioned in Figure 12 refers to mix.

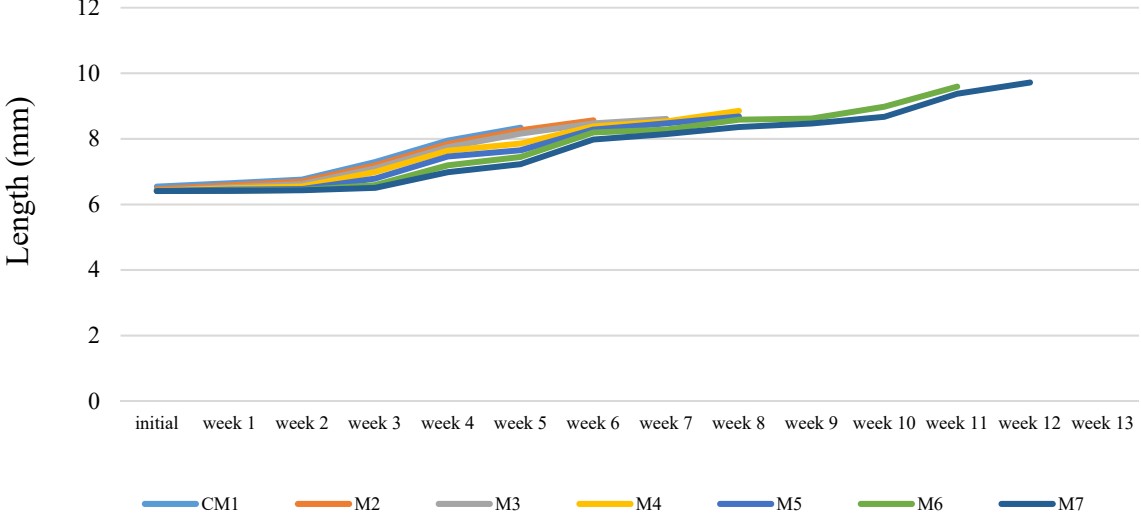

**Figure 12.** Sulphate attack of aerated concrete.

## 4. Conclusions

This research concentrates on the use of the agricultural waste product in the form of RHA as a cement substitute for aerated concert production and the conclusions follow.

The replacement of cement with RHA induced a density increase up to 10 percent and began to decrease by 12.5 percent.

The compressive strength, split tensile strength, and flexural strength results increase up to 10 percent of RHA and start to decrease at 12.5 percent. Maximum compressive strength, split tensile strength, and flexural strength were observed at 10% replacement of cement with RHA, with results of 22.16%, 20.41%, and 22.31% greater than those of the control mix at 28 days, respectively.

Corrosion potential values decrease with the increasing RHA percentage which helps in reducing corrosion.

Sulphate attack results show that with an increase in RHA percentage, the effect of sulphate decreases.

Based on the conducted research, it can be concluded that a 10 percent replacement of cement with RHA is optimum.

It can be suggested from the conducted study that when using RHA as a supplementary material, dual-fold benefits can be achieved. The first is to reduce the overall cost as 10 percent RHA replacement shows compatible results and secondly, the use of RHA plays an important role in maintaining an eco-friendly environment.

**Author Contributions:** T.A., A.S. & D.K.B. designed the study; T.A. performed the experimental work and analyzed the data; A.S., Z.A., A.S.B. & D.K.B. reviewed the paper; T.A. & A.S. wrote the paper. All authors have read and agreed to the published version of the manuscript.

**Funding:** All authors certify that they have no affiliations with or involvement in any organization or entity with any financial interest or non-financial interest in the subject matter or materials discussed in this manuscript.

**Data Availability Statement:** The data presented in this study are available on request from the corresponding author.

**Acknowledgments:** Not applicable.

**Conflicts of Interest:** The authors declare no conflict of interest.

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
