# Peer review of "Mechanical and Durability Properties of Aerated Concrete Incorporating Rice Husk Ash (RHA) as Partial Replacement of Cement"

_crystals, doi:10.3390/cryst11060604_

Round 1
Reviewer 1 Report
In this paper, a small amount (~10%) of the RHA was used replacing cement in the manufacturing fo the aerated concrete. And the RHA improve the mechanical and durability properties of the aerated concrete. However, some quality improvements are needed and the information provided should be supplemented. There are several questions for the authors. 1. The difference between the previously research and this paper is that RHA is applied to 'aerated concrete'. However, there is no accurate information on aerated concrete (ex. Porosity, void size etc.) It should be supplemented. Add a cross-section images of the manufactured aerated concretes to show the structure of the aerated concrete. 2. It is difficult to identify the information (~high specific surface area?) author want to convey in Figures 1 and 2. RHA dimensions, structure and etc. should be indicated more clearly in the Figures 1 and 2. 3. What are the standards for the mechanical tests (Compressive, split tensile, and flexural strength). The dimensions, geometry and test conditions must be added to manuscript. The authors could include a figure with the dimensions and geometry of the specimens. 4. There is no deviation bar in the mechanical test results. How many times did the experiment repeat for each condition? Please add the divation to mechanical test results. 5. Add actual images of fabricated and fractured specimens.
Author Response
The authors are very much thankful to the reviewer for their precious time and for giving valuable suggestions for the improvement of the manuscript.

Reviewer 2 Report
This paper describes mechanical and durability properties of aerated concrete incorporating rice husk ash as partial replacement of cement. The topic of the paper, in general, fits well within Crystals. However, the manuscript is very descriptive, and is essentially an analytical report, drawing only a weak (if any) interpretation of results obtained. The report style of the article makes it hard to judge what is the novelty or true scientific meaning of this work. This could be improved by a better structure and presenting the interpretation-evaluation the findings of this paper.
1. The novelty of the paper should be stressed: what is really new with respect to the state of the art. As authors themselves highlight, the use of rice husk ash as partial replacement of cement is a common practice worldwide.
2. What is the shortage of the knowledge in the literature? You should address it in Introduction. The authors should provide a citation for each of these supposedly outstanding properties. Similar citations in [4–7] should be given more information for each of the reference.
3.What is the reliability of the results? Have the authors duplicated all the test runs? The authors need to address all the analysis items in details. It's important for readers to understand how the experiments be done. Please specify the experimental procedures more details.
4. The description of the choice of the w/c=0.6 is not clear. Please provide the reason.
5.Some explanations are too speculative. Please provide some theories or compare the results with previous researches.
6. The relative perspectives and suggestions to the ultimate goal of this paper should be addressed in the conclusion.
Author Response
The authors are very much thankful for the reviewers for their precious time and giving valuable comments for the improvement of the manuscript

Round 2
Reviewer 1 Report
All questions and comments were clearly revised.
Reviewer 2 Report
I appreciate the author's effort for incorporating all suggestions. I agree this paper for publication. Thank you very much.